# Behavior Transformers:
# Cloning $k$ modes with one stone

**Nur Muhammad (Mahi) Shafiullah** *

**Zichen Jeff Cui**

**Ariuntuya Altanzaya**

**Lerrel Pinto**

**New York University**

## Abstract

While behavior learning has made impressive progress in recent times, it lags behind computer vision and natural language processing due to its inability to leverage large, human-generated datasets. Human behaviors have wide variance, multiple modes, and human demonstrations typically do not come with reward labels. These properties limit the applicability of current methods in Offline RL and Behavioral Cloning to learn from large, pre-collected datasets. In this work, we present Behavior Transformer (BeT), a new technique to model unlabeled demonstration data with multiple modes. BeT retrofits standard transformer architectures with action discretization coupled with a multi-task action correction inspired by offset prediction in object detection. This allows us to leverage the multi-modal modeling ability of modern transformers to predict multi-modal continuous actions. We experimentally evaluate BeT on a variety of robotic manipulation and self-driving behavior datasets. We show that BeT significantly improves over prior state-of-the-art work on solving demonstrated tasks while capturing the major modes present in the pre-collected datasets. Finally, through an extensive ablation study, we analyze the importance of every crucial component in BeT. Videos of behavior generated by BeT are available here: `https://mahis.life/bet`.

## 1 Introduction

Creating agents that can behave intelligently in complex environments has been a longstanding problem in machine learning. Although Reinforcement Learning (RL) has made significant advances in behavior learning, its success comes at the cost of high sample complexity [57, 23, 1]. Without priors on how to behave, state-of-the-art RL methods require online interactions on the order of 1-10M 'reward-labeled' samples for benchmark control tasks [81]. This is in stark contrast to vision and language tasks, where pretrained models and data-driven priors are the norm [19, 11, 32, 6], which allows for efficient downstream task solving.

So how do we learn behavioral priors from pre-collected data? One option is offline RL [47], where offline datasets coupled with conservative policy optimization can learn task-specific behaviors. However, such methods have yet to tackle domains where task-specific reward labels are not present. Without explicit reward labels, imitation learning, particularly behavior cloning, is a more fitting option [68, 9, 77]. Here, given behavior data $\mathcal{D} \equiv \{s_t, a_t\}$, behavior models can be trained to predict actions $f_\theta(s_t) \rightarrow a_t$ through supervised learning. When demonstration data is plentiful, such approaches have found impressive success in a variety of domains from self-driving [68, 14] to robotic manipulation [84, 62]. Importantly, it requires neither online interactions nor reward labels.

---

*Corresponding author, email: `mahi@cs.nyu.edu`

36th Conference on Neural Information Processing Systems (NeurIPS 2022).

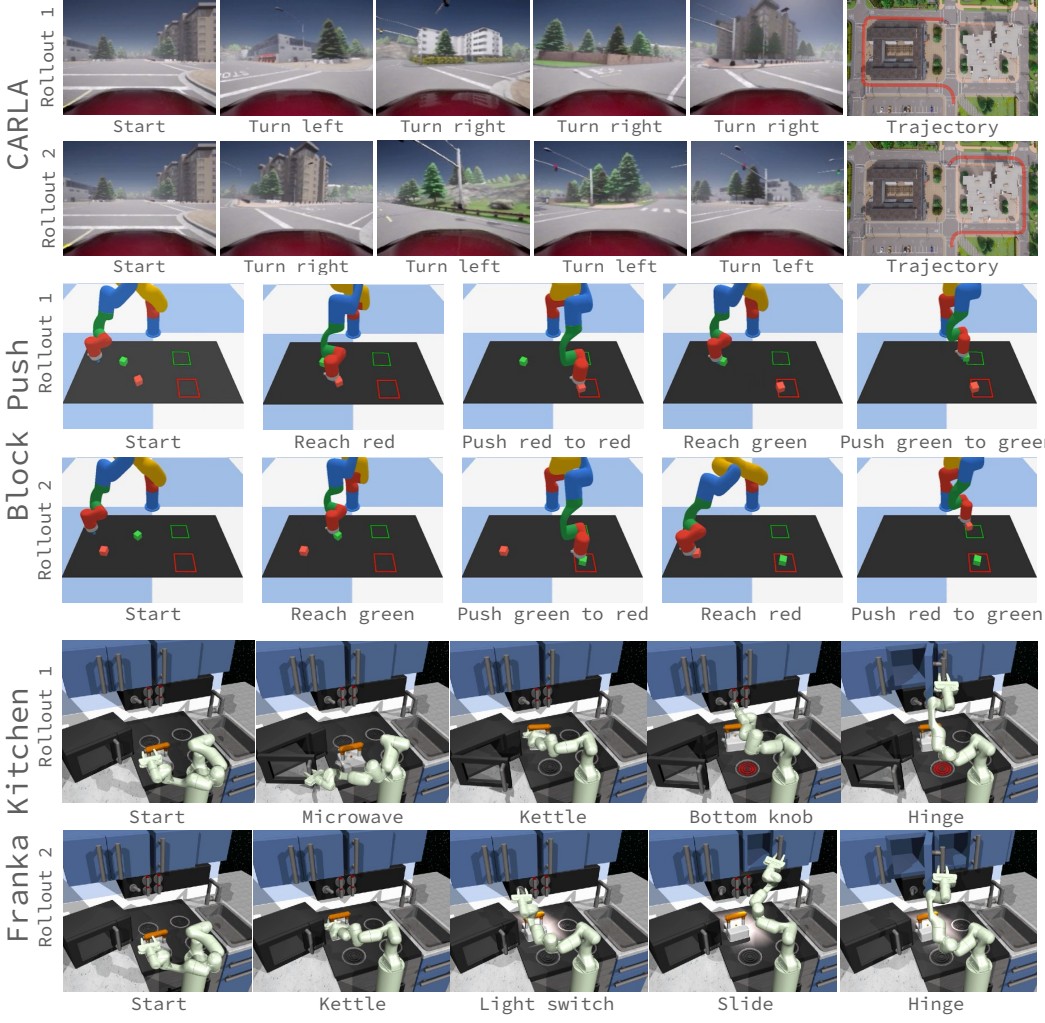

Figure 1: Unconditional rollouts from BeT models trained from multi-modal demonstartions on the CARLA, Block push, and Franka Kitchen environments. Due to the multi-modal architecture of BeT, even in the same environment successive rollouts can achieve different goals or the same goals in different ways.

However, state-of-the-art behavior cloning methods often make a fundamental assumption – that the data is drawn from a unimodal expert solving a single task. This assumption is often baked in to the architecture design, such as using a Gaussian prior. On the other hand, natural pre-collected data is sub-optimal, noisy, and contains multiple modes of behavior, all entangled in a single dataset. This distributionally multi-modal experience is most prominent in human demonstrations. Not only do we perform a large variety of behaviors every day, our personal biases result in significant multi-modality even for the same behavior [31, 52]. Current approach for behavior cloning from such datasets primarily focus on learning goal-conditioned policies, where each goal implies a single mode of behavior [35, 34, 52, 16]. However, even after goal-conditioning, an important question remains: How do we train models that can natively "clone" multi-modal behavior data?

In this work, we present Behavior Transformers (BeT), a new method for learning behaviors from rich, distributionally multi-modal data. BeT is based of three key insights. First, we leverage the context based multi-token prediction ability of transformer-based sequence models [78] to predict multi-modal actions. Second, since transformer-based sequence models are naturally suited to predicting discrete classes, we cluster continuous actions into discrete bins using k-means [53]. This allows us to model high-dimensional, continuous multi-modal action distributions as categorical distributions without learning complicated generative models [42, 20]. Third, to ensure that the actions sampled from BeT are useful for online rollouts, we concurrently learn a residual action corrector to produce continuous actions for a sampled action bin.

We experimentally evaluate BeT on five datasets ranging from simple diagnostic toy datasets to complex datasets that include simulated robotic pushing [25], sequential task solving in kitchen environments [34], and self-driving with visual observations in CARLA [21]. The two main findings from these experiments can be summarized as:

1. On multi-modal datasets, BeT achieves significantly higher performance during online rollouts compared to prior behavior modelling methods.

2. Rather than collapsing or latching onto one mode, BeT is able to cover the major modes present in the training behavior datasets. Unconditional rollouts from this model can be seen in Fig. 1.

All of our datasets, code, and trained models will be made publicly available.

## 2    Behavior Transformers

Given a dataset of continuous observation and action pairs $\mathcal{D} \equiv \{(o, a)\} \subset \mathcal{O} \times \mathcal{A}$ that contains behaviors we are interested in, our goal is to learn a behavior policy $\pi : \mathcal{O} \mapsto \mathcal{A}$ that models this data without any online interactions with the environment or reward labels. This setup follows the Behavior Cloning formulation, where policies are trained to model demonstrations from expert rollouts. Often, such policies are chosen from a hypothesis class parametrized by parameter set $\theta$. Following this convention, our objective is to find the parameter $\theta$ that maximizes the probability of the observed data

$$\theta^* := \arg\max_{\theta} \prod_t \mathbb{P}(a_t \mid o_t; \theta) \tag{1}$$

When the model class is restricted to unimodal isotropic Gaussians, this maximum likelihood estimation problem leads to minimizing the Mean Squared Error (MSE), $\sum_t \|a_t - \pi(o_t; \theta)\|^2$.

**Limitations of traditional MSE-based BC:**   While MSE-based BC has been able to solve a variety of tasks [9, 77], it assumes that the data distribution is unimodal. Clean data from an expert demonstrator solving a particular task in a particular way satisfies this assumption, but pre-collected intelligent behavior often may not [52, 34]. While more recent behavior generation models have sought to address this problem, they often require complex generative mod-

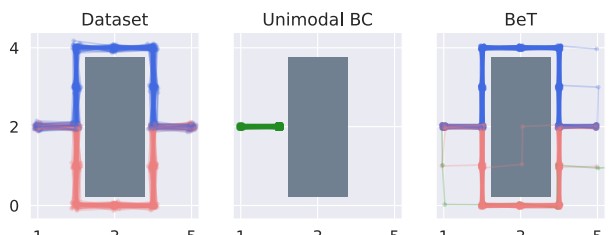

Figure 2: Comparison between a regular MSE-based BC model and a BeT models that can capture multi-modal distributions. The MSE-BC model takes 0 action to minimize MSE.

els [76], an exponential number of bins for actions [54], complicated training schemes [66], or time-consuming test-time optimization [25]. An experimental analysis of some of these prior works is presented in Section 3.

**Overview of Behavior Transformers (BeT):**   We address two critical assumptions in regular BC. First, we relax the assumption that the behavior we are cloning is purely Markovian, and instead model $P(a_t \mid o_t, o_{t-1}, \cdots, o_{t-h+1})$ for some horizon $h$. Second, instead of assuming that actions are generated by a unimodal action distribution, we model our action distribution as a mixture of gaussians. However, unlike previous efforts similar to Mixture Density Networks (MDN) to do so, whose limitations have been explored in Florence et al. [25], we do not explicitly predict mode centers, which significantly improves our modeling capacity. To operationalize these two features in a single behavior model, we make use of transformers since (a) they are effective in utilizing prior observational history, and (b) they are naturally suited to output multi-modal tokens through their architecture.

### 2.1    Action discretization for distribution learning

Although transformers have become standard as a backbone for sequence-to-sequence models [19, 11], they are designed to process discrete tokens and not continuous values. In fact, modeling multi-modal

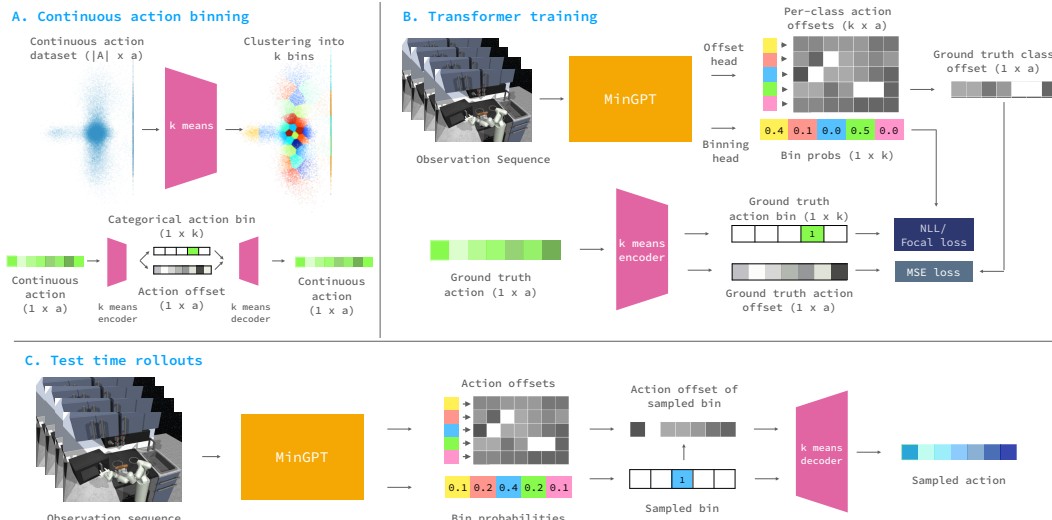

Figure 3: Architecture of Behavior Transformer. (A) The continuous action binning using k-means algorithm that lets BeT split every action into a discrete bin and a continuous offset, and later combine them into one full action. (B) Training BeT using demonstrations offline; each ground truth action provides a ground truth bin and residual action, which is used to train the minGPT trunk with its binning and action offset heads. (C) Rollouts from BeT in test time, where it first chooses a bin and then picks the corresponding offset to reconstruct a continuous action.

distributions of high-dimensional continuous variables in a tractable manner is in itself a challenging problem, especially if we want the trained behavior model to cover the modes present in the dataset. To address this, we propose a new factoring of the action prediction task by dividing each action in two parts: a categorical variable denoting an 'action center', and a corresponding 'residual action'.

To this end, given the actions in our dataset, we first optimize for a set of $k$ action centers, $\{A_1, A_2, \cdots, A_k\} \subset \mathcal{A}$. We then decompose each action into two parts: a categorical variable representing the closest action bin, $\lfloor a \rfloor := \arg\min_i \|a - A_i\|_2$, and a continuous residual action $\langle a \rangle := a - A_{\lfloor a \rfloor}$. If we are given the set of action centers $\{A_i\}_{i=1}^k$, an action bin index $\lfloor a \rfloor$ and the residual action $\langle a \rangle$, we can deterministically reconstruct the true action $a := A_{\lfloor a \rfloor} + \langle a \rangle$. Once learned, these k-means based encoder and decoders for this action factorization process are fixed for the rest of the train and testing phases. The action factorization procedure is illustrated in Fig. 3 (A).

## 2.2 Attention-based behavior mode learning

Once we have the clustering based autoencoder learned from the actions in the dataset, we model our demonstration trajectories with BeT. We use a transformer decoder model, namely minGPT [11], with minor modifications, as our backbone. The transformer $\mathcal{T}$ takes in a sequence of continuous observations $(o_i, o_{i+1}, \cdots, o_{i+h-1})$ and learns a sequence-to-sequence model mapping each observation to a categorical distribution over $k$ discrete action bins. The predicted probability sequence is then compared with the ground truth labels, $(\lfloor a_i \rfloor, \lfloor a_{i+1} \rfloor, \lfloor a_{i+2} \rfloor, \cdots, \lfloor a_{i+h-1} \rfloor)$. We use a negative log-likelihood-based Focal loss [49] between the predicted categorical distribution probabilities and the ground truth labels to train the transformer head. Focal loss is a simple modification over the standard cross entropy loss. While the standard cross entropy loss for binary classification can be thought of $\mathcal{L}_{ce}(p_t) = -\log(p_t)$, Focal loss adds a term $(1 - p_t)^\gamma$ to this, to make the new loss

$$\mathcal{L}_{focal}(p_t) = -(1 - p_t)^\gamma \log(p_t)$$

This loss has the interesting property that its gradient is more steep for smaller values of $p_t$, while flatter for larger values of $p_t$. Thus, it penalizes and changes the model more for making errors in the low-probability classes, while is more lenient about making errors in the high probability classes. The model is illustrated in Fig. 3 (B).

## 2.3 Action correction: from coarse to finer-grained predictions

Using a transformer allows us to model multi-modal actions. However, discretizing the continuous action space in any way invariably causes loss of fidelity [39]. Discretization error may cause online rollouts of the behavior policy to go out of distribution from the original dataset [73], which can in turn cause critical failures. To predict the complete continuous action, we add an extra head to the transformer decoder that offsets the discretized action centers based on the observations.

For each observation $o_i$ in the sequence, the head produces a $k \times \dim(A)$ matrix with $k$ proposed residual action vectors, $\left( \langle a_i^{(j)} \rangle \right)_{j=1}^{k} = (\langle \hat{a}_i^{(1)} \rangle, \langle \hat{a}_i^{(2)} \rangle, \langle \hat{a}_i^{(3)} \rangle, \cdots, \langle \hat{a}_i^{(k)} \rangle)$, where the residual actions correspond to bin centers $A_1, A_2, A_3, \cdots, A_k$. These residual actions are trained with a loss akin to the *masked multi-task loss* [30] from object detection. In our case, if the ground truth action is $\mathbf{a}$, the loss is:

$$\text{MT-Loss}\left(\mathbf{a}, \left(\langle \hat{a}_i^{(j)} \rangle\right)_{j=1}^{k}\right) = \sum_{j=1}^{k} \mathbb{I}[\lfloor \mathbf{a} \rfloor = j] \cdot \| \langle \mathbf{a} \rangle - \langle \hat{a}^{(j)} \rangle \|_2^2 \tag{2}$$

Where $\mathbb{I}[]$ denotes the Iverson bracket, ensuring the offset head of BeT only incurs loss from the ground truth class of action $\mathbf{a}$. This mechanism prevents the model from trying to fit the ground truth action using the offset at every index.

## 2.4 Test-time sampling from BeT

During test time, at timestep $t$ we input the latest $h$ observations $(o_t, o_{t-1}, \cdots, o_{t-h+1})$ to the transformer, combining the present observation $o_t$ with $h-1$ previous observations. Our trained MinGPT model gives us $h \times 1 \times k$ bin center probability vectors, and $h \times k \times \dim(A)$ offset matrix. To sample an action at timestep $t$, we first sample an action center according to the predicted bin center probabilities on the $t^{\text{th}}$ index. Once we have chosen an action center $A_{t,j}$, we add the corresponding residual action $\langle \hat{a}_t^{(j)} \rangle$ to it to recover a predicted continuous action $\hat{\mathbf{a}_t} = A_{t,j} + \langle \hat{a}_t^{(j)} \rangle$. This sampling procedure is illustrated in Fig. 3 (C).

# 3 Experiments

We now study the empirical performance of BeT on a variety of behavior learning tasks. Our experiments are designed to answer the following questions: (a) Is BeT able to imitate multi-modal demonstrations? (b) How well does BeT capture the modes present in behavior data? (c) How important are the individual components of BeT?

## 3.1 Environments and datasets

We experiment with five broad environments. While full descriptions of these environments, dataset creation procedure, and overall statistics are in Appendix A, a brief description of them are as follows.

(a) **Point mass environment #1:** Our first set of experiments in Fig. 2, used to get a qualitative understanding of BeT, were performed in a simple Pointmass environment with a 2D observation and action space with two hundred demonstrations. The pre-collected demonstrations start at a fixed point, and then make their way to another point while avoiding a block in the middle. The two primary modes in this dataset are taking a left turn versus a right turn.

(b) **Point mass environment #2:** The setup is similar to the previous environment with the exception of one straight line and two complicated prolonged 'Z' shaped modes of demonstration (Fig. 5.)

(c) **CARLA self-driving environment:** CARLA [21] uses the Unreal Engine to provide a simulated driving environment in a visually realistic landscape. The agent action space is 2D (accelerate/brake and left/right steer), while the observation space is (224,224,3)-dimensional RGB image from the car. A hundred total demonstrations drive around a building block in two distinct modes. This environment highlights the challenge of behavior learning from high-dimensional observations as shown in Fig. 1 (a). For visual observations with BeT, we use a frozen ResNet-18 [36] pretrained on ImageNet [18] as an encoder.

(d) **Multi-modal block-pushing environment:** For more complicated interaction data, we use the multi-modal block-pushing environment from Implicit Behavioral Cloning (IBC) [25], where an XArm robot needs to push two blocks into two squares in any order. The blocks and target squares are colored red and green. The positions of the blocks are randomized at episode start. We collect 1,000 demonstrations using a deterministic controller with two independent axes of multi-modality: (a) it starts by reaching for either the red or the green block, with 50% probability, and (b) it pushes the blocks to (red, green) or (green, red) squares respectively with 50% probability.

(e) **Franka kitchen environment:** To highlight the complexity of performing long sequences of actions, we use the Relay Kitchen Environment [34] where a Franka robot manipulates a virtual kitchen environment. We use the relay policy learning dataset with 566 demonstrations collected by human participants wearing VR headsets. The participants completed a sequence of four object-interaction tasks in each episode [34]. There are a total of seven interactable objects in the kitchen: a microwave, a kettle, a slide cabinet, a hinge cabinet, a light switch, and two burner knobs. This dataset contains two different kinds of multi-modality: one from the inherent noise in human demonstrations, and another from the demonstrators' intent.

## 3.2 Baseline behavior learning methods

While a full description of our baselines are in Appendix B.1, a brief description of them is here:

(a) **Multi-layer Perceptron with MSE (RBC):** We use MLP networks trained with MSE loss as our first baseline, since this is the standard way of performing behavioral cloning for a new task [77]. A comparison with transformer-based behavior cloning is discussed in Section 3.5.

(b) **Nearest neighbor (NN):** Nearest neighbor based algorithms are easy to implement, and has recently shown to have strong performance on complicated behavioral cloning tasks [3].

(c) **Locally Weighted Regression (LWR):** This non-parametric approach provides better regularization compared to NN and is a strong alternative to parametric BC [4, 62].

(d) **Variational auto-encoders (VAE):** Inspired by SPiRL [66], where behavioral priors are learned through a VAE [42], we compare with continuous actions generated from the VAE and the prior.

(e) **Normalizing Flow (Flow):** Inspired by PARROT [76], where state-conditioned action priors are learned through a Flow model [20], we compare with actions generated from the Flow model.

(f) **Implicit Behavioral Cloning (IBC):** Instead of modeling the conditional distribution $P(a \mid o)$, IBC models the joint probability distribution $P(a, o)$ using energy-based models [25]. While IBC is slower than explicit BC models because of their sampling requirements, they have been shown to learn well on multi-modal data, and outperform earlier work such as MLP-MDNs [8].

## 3.3 Is BeT able to imitate multi-modal demonstrations?

The first question we ask is whether BeT can actually clone behaviors given a mixed dataset of unlabeled, multi-modal behaviors. To examine that, we look at the performance of our model in CARLA, Block push, and Kitchen environments compared with our baselines in Table 1.

We see that BeT outperforms all other methods in all environments except CARLA, where it is narrowly outperformed by LWR. Since the models are all behavioral cloning algorithms, they share the failure mode of failing once the observations go out of distribution (OOD). However, they vary in the tolerance. For example, BeT shines in the Block push environment, where alongside extreme environment randomness and multi-modality, the models also have to learn significant long-term behaviors and commit to a single mode over a long period. While all baselines can somewhat successfully reach one block, they fail to complete the long-horizon, multi-modal task of pushing two blocks into two different bins. On the other hand, we observe that BeT's primary failure mode is not realizing a block has not completely entered the target yet, while other methods either go OOD quickly, or keep switching between modes. We also observe that BeT performs well even in complex observation and action spaces. In the CARLA environment, the model takes in visual observations, while in the Franka Kitchen environment, the action space corresponds to a 9-DOF torque controlled robot. BeT handles both cases with the same ease as it does environments with lower-dimensional observation or action spaces.

Table 1: Performance of BeT compared with different baselines in learning from demonstrations. For CARLA, we measure the probability of the car reaching the goal successfully. For Block push, we measure the probability of reaching one and two blocks, and the probabilities of pushing one and two blocks to respective squares. For Kitchen, we measure the probability of $n$ tasks being completed by the model within the allotted 280 timesteps. Evaluations are over 100 rollouts in CARLA and 1,000 rollouts in Block push and Kitchen environments.

| | CARLA | Block push | | | | Kitchen | | | | |
| | Driving | Reach | | Push | | # Tasks completed | | | | |
| Baselines | Success | R1 | R2 | P1 | P2 | 1 | 2 | 3 | 4 | 5 |
|---|---|---|---|---|---|---|---|---|---|---|
| RBC | 0.98 | 0.67 | 0 | 0 | 0 | 0 | 0 | 0 | 0 | 0 |
| 1-NN | 0.99 | 0.49 | 0.05 | 0.01 | 0 | 0.90 | 0.72 | 0.44 | 0.17 | 0 |
| LWR | **1** | 0.50 | 0.06 | 0 | 0 | **1** | 0.83 | 0.52 | 0.21 | 0 |
| VAE | 0 | 0.60 | 0.05 | 0 | 0 | **1** | 0 | 0 | 0 | 0 |
| Flow | 0.03 | 0.59 | 0.02 | 0 | 0 | 0.04 | 0 | 0 | 0 | 0 |
| IBC | 0.25 | 0.98 | 0.04 | 0.01 | 0 | 0.99 | 0.87 | 0.61 | 0.24 | 0 |
| BeT (Ours) | 0.98 | **1** | **0.99** | **0.96** | **0.71** | 0.99 | **0.93** | **0.71** | **0.44** | **0.02** |

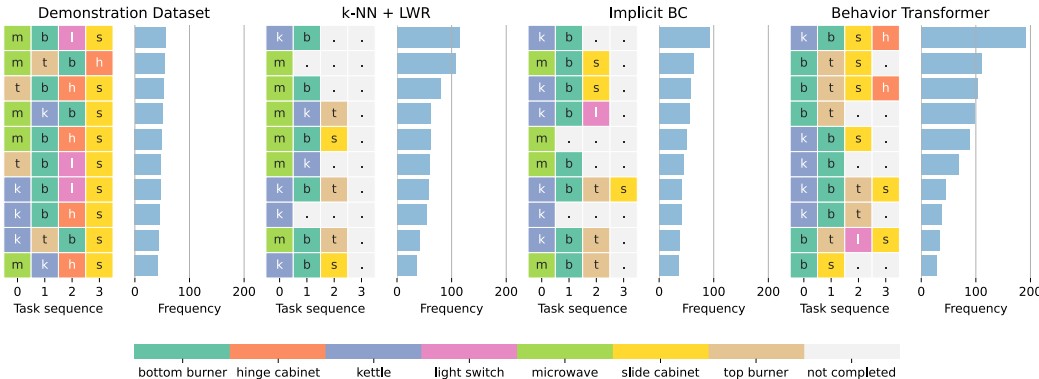

bottom burner   hinge cabinet   kettle   light switch   microwave   slide cabinet   top burner   not completed

Figure 4: Distribution of most frequent tasks completed in sequence in the Kitchen environment. Each task is colored differently, and frequency is shown out of a 1,000 unconditional rollouts from the models.

## 3.4 Does BeT capture the modes present in behavior data?

Next, we examine the question of whether, given a dataset where multi-modal behavior exists, our model learns behavior that is also multi-modal. Here, we are interested in seeing the variance of the behavior of the model over different rollouts. In each of our environments, the demonstrations contain different types of multi-modality. As a result, we show a comprehensive analysis of multi-modality seen in our agent behaviors.

Table 2: Multimodality learned from the multimodal demonstrations by different algorithms. In CARLA, we consider the probability of turning left vs. right at the intersection, ignoring OOD rollouts. In Block push, we consider two set of probabilities, (a) which block was reached first, and (b) what was the pushing target for each block. Finally, in Franka Kitchen, we consider the empirical entropy for the task sequences, considered as strings, sampled from the model. We highlight the values closest to the corresponding demonstration values.

| | CARLA | | Block: first block reached | | Push: red block target | | Push: green block target | | Kitchen |
| Baselines | Left | Right | Red | Green | Red | Green | Red | Green | Task entropy |
|---|---|---|---|---|---|---|---|---|---|
| RBC | 0 | 0.98 | 0.41 | 0.25 | 0 | 0 | 0 | 0 | 0 |
| 1-NN | 0 | 0.99 | 0.24 | 0.25 | 0 | 0 | 0 | 0.01 | 2.12 |
| LWR | 0 | 1 | 0.26 | 0.26 | 0.01 | 0 | 0.01 | 0.01 | 2.29 |
| VAE | 0 | 0 | 0.27 | 0.33 | 0 | 0 | 0 | 0 | 0.72 |
| Flow | 0 | 0 | 0.31 | 0.29 | 0 | 0 | 0 | 0 | 0.08 |
| IBC | 0.12 | 0.13 | **0.48** | **0.50** | 0 | 0 | 0.01 | 0.01 | 2.41 |
| BeT (Ours) | **0.34** | **0.64** | 0.54 | 0.46 | **0.43** | **0.44** | **0.41** | **0.40** | **2.47** |
| Demonstrations | 0.50 | 0.50 | 0.50 | 0.50 | 0.50 | 0.50 | 0.50 | 0.50 | 2.96 |

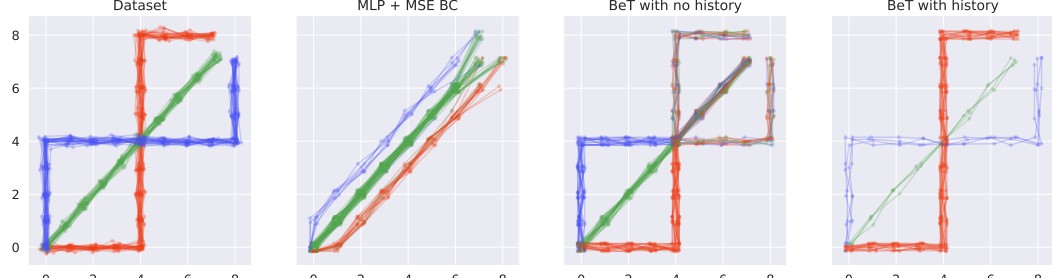

Figure 5: Comparison between an RBC model and two BeT models, trained with and without historical context on a dataset with three distinct modes. BeT with history is better able to capture the context-dependant behavior in the demonstrations.

We see in Table 2 that in CARLA and Block push, BeT covers all the modes of the demonstration data, even in the few cases where it does not perfectly match the demonstrated task probabilities. For the Kitchen environment, we see in Fig. 4 that BeT visits certain strings of tasks more frequently than in the original demonstrations. However, compared to other strong baselines, BeT generates longer task strings more often while maintaining diversity and not collapsing to a single mode.

### 3.5 How important are the individual components of BeT?

There are four key differences between BeT architecture and standard BC: (a) binning actions into discrete clusters, (b) using offsets to faithfully reconstruct actions later, (c) learning sequentially to use historical context, and (d) using an attention-based MinGPT trunk. In this section, we discuss the impacts they have in BeT performance.

**Impact of discrete binning:** Intuitively, having discrete options for bin centers is what enables BeT to express multi-modal behavior even when starting from an identical starting state. Indeed, if there is no binning, we see from Table 3 that the performance of BeT drops significantly. More tellingly, in the Franka Kitchen environment, the model only ever completed a subsequence of (kettle, top/bottom burner, light switch, slide cabinet) tasks after 100 random rollouts. This result shows us that having discrete bins helps BeT achieve multi-modality.

Table 3: Relative performance of ablated variants of BeT, normalized by average BeT successes at the task

| Ablations | CARLA | Block push | Kitchen |
|---|---|---|---|
| No offsets | 0.94 | 0.95 | 0.78 |
| No binning | 0.94 | 0.25 | 0.68 |
| No history | 0.65 | 0.95 | 0.88 |
| MLP | 0.90 | 0 | 0.05 |
| Temp. Conv | 0.72 | 0.01 | 0.26 |
| LSTM | 0.03 | 0.03 | 0.04 |
| GPT-MDN | 0.30 | 0.83 | 0.86 |
| Unif. quant. | 0.90 | 0.96 | 0.90 |

We also experiment with the Mixture density networks (MDN) [8] and uniform quantization, as shown in previous works [25, 39]. We see that they may perform well sometimes but overall still fall short of our k-means binning approach.

**Necessity of action offsets:** An important feature of BeT is the residual action offset that corrects the discrete actions coming from the bins. While the bin centers may be quite expressive, Table 3 shows that the inability to correct them causes a performance degrade. Interestingly, the largest degradation comes in the Kitchen environment, which also has the highest dimensional action space. Intuitively, we can understand how in higher dimension the loss of fidelity from discretizing would be higher, and the relative performance loss across three environments support that hypothesis.

**Importance of historical context:** While RL algorithms traditionally assume environments are Markovian, human behavior in an open-ended environment is rarely so. Thus, using historical context helps BeT to perform well. We show a simple experiment in Fig. 5 on the second point mass environment. Here, training and evaluating with some historical context allows BeT to follow the demonstrations better. We experience the same in the CARLA, Block push, and Kitchen environments, where training with some historical context raises performance across the board as seen in Table 3.

**Importance of transformer architecture:** Despite transformers' success in other fields of machine learning, it is natural to wonder whether the tasks BeT solves here really requires one. We ablated

BeT by replacing the MinGPT trunk with an MLP, Temporal Convolution, and LSTMs, and found that they have lower performance while also being difficult to train stably. This performance reduction remains even if the MLP is given some historical context by stacking $h$ observations before passing it to the MLP. See Table. 3 for results and Appendix C.2 for further details.

**Ablating the number of discrete bin centers, $k$:**     Since BeT is trained with a sum of focal loss for the binning head and MSE loss for the offset head, the number of cluster centers present a trade-off in the architecture. Concretely, as the number of bins go up, the log-likelihood loss goes up but the MSE loss goes down. In Table 3, we showed that using only one bin ($k = 1$) decreases the performance level of BeT.

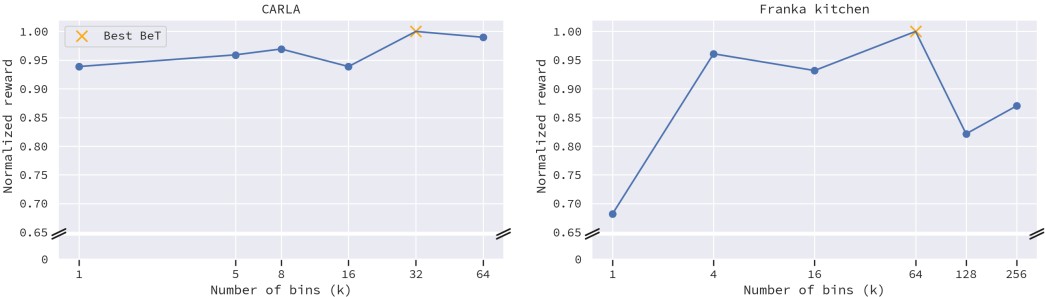

Figure 6: Ablating the number of discrete bin centers $k$ for BeT. Reward is normalized with respect to the best performing model.

In Fig. 6, we present the plot of the variation in performance as $k$ value changes. We see that for BeT it matters somewhat to pick a number of clusters $k$ that is in the right neighborhood. However, the range for near-optimum performance is quite wide. In our experiments, we also pick a $k$ in the right neighborhood and only run a sweep at the very end to find out an optimal value for $k$.

**Computation considerations:**     While transformers in usual contexts are large models, we downscale them for our application in BeT (See Appendix B.4). Our models contain on the order of $10^4$–$10^6$ parameters, and even with a small batch size trains within an hour for our largest datasets (Block push) on a single desktop GPU. In contrast, for the same task, our strongest baseline IBC takes about 14 hours. Evaluation rollouts on the same environment take 1.65 seconds with BeT, as opposed to 17.70 seconds with IBC.

## 4   Related Work

This paper builds upon a rich literature in imitation learning, offline learning, generative models, and transformer architectures. The most relevant ones to our work are discussed here.

**Learning from offline data:**     Since Pomerleau [67] showed the possibility of driving an autonomous vehicle using offline data and a neural network, learning behavior from offline data has been a continuous topic of research for scalable behavior learning [2, 7, 75]. The approaches can be divided into two broad classes: Offline RL [27, 44, 45, 80, 47, 26], focusing on learning from datasets of a mixed quality that also have reward labels; and imitation learning [61, 63, 64, 37], focusing on learning behavior from a dataset of expert behavior without reward labels. BeT falls under the second category, as it is a behavior cloning model. Behavior cloning is a form of imitation learning that tries to model the action of the expert given the observation which is often used in real-world applications [84, 85, 84, 69, 24, 83]. As behavior cloning algorithms are generally solving a fully supervised learning problem, they tend to be faster and simpler than reinforcement learning or offline RL algorithms and in some cases show competitive results [26, 33].

**Generative models for behavior learning:**     One approach for imitation learning is Inverse Reinforcement Learning or IRL [74, 59], where given expert demonstrations, a model tries to construct the reward function. This reward function is then used to generate desirable behavior. GAIL [37], an IRL algorithm, connects generative adversarial models with imitation learning to construct a model that

can generate expert-like behavior. Under this IRL framework, previous works have tried to predict multi-modal, multi-human trajectories [46, 38]. Similarly, other works have tried Gaussian Processes [71] for creating dynamical models for human motion [79]. Another class of algorithms learn a generative action decoder [65, 52, 76] from interaction data to make downstream reinforcement learning faster and easier, which inspired BeT's action factorization. Finally, a class of algorithms, most notably [50, 25, 43, 58] do not directly learn a generative model but instead learn energy based models. These energy based models can then be sampled to generate desired behavior. Since [25] is a BC model capable of multi-modality, we compare against it as a baseline in Sec. 3.

**Transformers for control:**    With the stellar success of transformer models [78] in natural language processing [19, 11] and computer vision [22], there has been significant interest in using transformer models to learn behavior and control. Among those, [12, 39] applies them to Reinforcement Learning and Offline Reinforcement Learning, respectively, while [13, 16, 54] use them for imitation learning. [16, 54] use transformers mostly to summarize historical visual context, while [13] relies on their long-term extrapolation abilities to collect human-in-the-loop demonstrations more efficiently. BeT is inspired by both of these use cases, as we use a transformer to summarize historical context while leveraging its generative abilities. Architecturally, BeT is most closely related to the imitation learning variant of [39], with a significant difference that while [39] learns the joint state, action distribution, BeT learns the conditional distribution of action given state, which allows BeT to tackle much more complicated state spaces.

**Datasets for distributionally multi-modal data:**    Similar to computer vision [18, 48, 51] and natural language processing [10, 70], there has been a recent interest in collecting behavior datasets that may aid in downstream behavior learning. Some of them are labeled with agent goals or rewards for downstream tasks [55, 26, 56], while others are more open ended [34, 52, 82] and come without reward or task labels. In our work, we focus towards the latter class. The lack of labeled goal or reward labels in the second category implies that there is more multi-modality in the action distributions compared to action distributions of goal or reward conditioned datasets, which is the same reason a lot of work learning from multi-modal datasets try to learn a goal-conditioned model [35, 34, 52, 16]. Finally, the lack of labelling requirements mean that the unlabelled datasets are cheaper to obtain, which should help BeT scale further in the future.

## 5   Discussions

In this work, we introduce Behavior Transformers (BeT), which uses a transformer-decoder based backbone with a discrete action mode predictor coupled with a continuous action offset corrector to model continuous actions sequences from open-ended, multi-modal demonstrations. While BeT shows promise, the truly exciting use of it would be to learn diverse behavior from human demonstrations or interactions in the real world. In parallel, extracting a particular, unimodal behavior policy from BeT during online interactions, either by distilling the model or by generating the right 'prompts' [72], would make BeT tremendously useful as a prior for online Reinforcement Learning.

## Acknowledgments

We thank Ben Evans, David Brandfonbrener, Abitha Thankaraj, Jyo Pari, and Anthony Chen for valuable feedback and discussions. This work was supported by grants from Honda, and ONR award numbers N00014-21-1-2404 and N00014-21-1-2758.

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
