# OpenReview forum: "Behavior Transformers: Cloning $k$ modes with one stone"
_NeurIPS.cc/2022/Conference — NeurIPS 2022 Accept_

### Official Review · Reviewer_uHYM · 2022-06-20

**Rating:** 8
**Confidence:** 5
**Soundness:** 3 good
**Presentation:** 3 good
**Contribution:** 3 good

**Summary:**

This paper describes a method for Behavior Cloning that leverages 1) a transformer-based architecture to model the state-action relationships, thereby lifting some of the Markovian assumptions many BC architectures make, and 2) a discrete+continuous residual action representation, which enables the model to learn a multimodal action representation more easily through binning of the discrete action clusters, while not incurring a large loss of precision by explicitly learning a continuous offset from the cluster centers.

**Questions:**

1- I am very curious about the choice of k-means as a discretization scheme vs. e.g. uniform quantization. Given the added correction term, it is not immediately obvious that performing k-means on the action is worth the added complexity. A uniform binning would also be more general and potentially would generalize better across tasks and embodiments. I would have loved to see that particular ablation.
2- The paper doesn't mention computational considerations at run-time, only at training time. I would have liked to see some discussion of the tradeoffs here, particularly if the model is to be used for real-time continuous control.

**Limitations:**

Nothing comes to mind.

**Strengths And Weaknesses:**

Strengths:
- the proposed architecture is well-motivated: much of recent works in BC have pointed at the weaknesses the model proposes to address, and the model architecture derives very naturally from the goals of addressing those weaknesses.
- the experimental results compare against strong baselines, and ablation experiments clearly tease out why the model performs well against them.
- the paper is limpid, well-organized and immediately useful to the community.

Weaknesses:
- No real-world experiment. This may not be essential in this context, but would inform how practical it is to deploy this kind of architecture in a real-time, real-world system.

---

> ### Author Response · Authors · 2022-08-02
> **Response to the Official Review of Paper729 by Reviewer uHYM**
>
> First of all, we thank you for your insightful review of the paper. We agree with you that the most exciting applications for algorithms such as BeT is in the real world, and would like to follow up this work with such applications in the future.
>
> Now, to answer your questions:
>
> 1. **K-means vs Uniform Quantization**: Firstly, we ran an ablation experiment with uniform quantization, added to table 3 now, which shows that while uniform quantization performs competitively on lower action dimension environments, its performance is below k-means on higher dimensions. We primarily chose k-means over uniform or quantile discretization for the same reason.
>
>     Uniform or quantile discretization per-dimension requires one of three things to happen: it needs an exponential blowup in the number of action bins, or an autoregressive action prediction causing quadratic blowup in transformer computation, or an assumption of independence between the action dimensions. Faced with such trade-offs, we instead chose k-means which only requires one extra matrix operation every step for discretizing or recovering actions. Complexity-wise, we believe that the extra overhead is worthwhile to keep our model small and fast. We believe the task of action discretization is worthy of additional research, as shown by concurrent research such as AQuaDem [1] presented in ICML 2022.
>
>
> 2. **Run-time computational consideration for BeT**: We currently present the run-time requirements of BeT compared with IBC in the computation considerations section, namely 1.65 second average runtime for each complete episode of Block-Push, as opposed to 17.70 seconds for IBC. In terms of raw computation time to determine one action from the observations, in the Kitchen environment, BeT took 2.8 ms, while IBC took 52 ms, Trajectory transformer took 867.86 ms, and a simple MLP, as the fastest point of comparison, took 0.5 ms..
>
>     At the same time, with our design choices, our run-time is asymptotically faster than possible transformer based baselines. Run-time computational expenses of a transformer depends on a few major points beyond the size of the transformer and the length of the historical context: in our case, linearly on the number of discrete bins, and quadratically on the number of tokens per actions or states. In BeT, using k-means discretization implies the number of bins can be much lower than uniform quantization. At the same time, treating one (state, action) pair as one input/output token pair yields an $O( (|S| + |A|)^2 )$ speedup over approaches like Trajectory Transformer, where each state action pair is encoded as $|S| + |A|$ input/output token pairs.
>
> We hope the above responses answer your questions sufficiently, and we thank again for your kind and positive feedback.
>
> [1] Dadashi, Robert, et al. "Continuous Control with Action Quantization from Demonstrations." arXiv preprint arXiv:2110.10149 (2021).

---

> > ### Comment · Reviewer_uHYM · 2022-08-03
> > **Thank you for the clarifications and additions**
> >
> > The overall discussion makes me more confident about my assessment (4->5)

---

### Official Review · Reviewer_xCyV · 2022-07-09

**Rating:** 7
**Confidence:** 4
**Soundness:** 3 good
**Presentation:** 4 excellent
**Contribution:** 2 fair

**Summary:**

The paper presents an approach to imitation learning using transformers. The core challenge this work aims to tackle is enabling imitation learning methods to better handle multiple behavior modes in demonstration data. The key idea amounts to learning a hybrid discrete+residual action representation using k means, then using a minGPT transformer architecture to produce actions conditioned on the history of observations and actions. The experiments show strong performance on tasks with multi-modality in the demonstrations compared to baselines, importance of each of the components, and some qualitative examples highlighting the methods ability to capture multimodality.

**Questions:**

- Can the authors explain the differences of their approach with the imitation learning version of trajectory transformer? A comparison would also be good to see.
- A more minor point: are there places where k-means fails is insufficient for binning the actions? I can imagine cases where small differences in the action can have huge impacts in task success, and perhaps k means may incorrectly group two functionally different actions to the same group. Perhaps there are ways of doing task-aware discretization of the actions.

**Limitations:**

Yes.

**Strengths And Weaknesses:**

*Strengths*
- The paper tackles an important problem in handling multi-modal behavior in demonstration data. Handling such data will be important for enabling imitation learning algorithms to scale to broader datasets.
- The proposed approach is clean, simple, and well motivated. Indeed sequence models seem to be able to capture very multi-modal distributions in text, and by discretizing actions it makes sense that the sequence model can also capture multi-modality in behavior.
- Experiments are also thorough, and compare the method to most of the relevant baselines (Implicit BC, Nearest Neighbors, BC) across multiple environments. The qualitative examples also nicely highlight how the method captures many modes in behavior, and how historical context is important for doing so.
- The ablations show that changing the architecture or removing the action binning/offset both hurt performance, suggesting that all of the components are important.
- Overall, the paper is well written.

*Weaknesses*
- The most significant weakness/question I have with this work is the novelty compared to the Trajectory Transformer (Janner et al). In the related work this paper simply states Trajectory Transformer as different because it focused on Offline RL. However, the Trajectory Transformer paper also proposes a version of the method that can be used for pure imitation learning (not just offline RL), and has corresponding experiments. Moreover, TT also tokenizes actions, and uses minGPT to generate sequences. The Trajectory Transformer paper proposes using beam search to guide actions toward certain goals, but without the beam search, the unconditional generation seems very similar to what is being proposed in BeT.  I can see some implementation differences like the masked MT loss and the k-means for discretization, but at a high level the methods appear very similar. At the very least, the current related work significantly underplays the similarities to TT, and I think a more comprehensive discussion of how the two are related is important. It should probably also be added as a comparison.

---

> ### Author Response · Authors · 2022-08-02
> **Response to Official Review of Paper729 by Reviewer xCyV**
>
> We thank you deeply for your thoughtful review. We have added a further baseline for Trajectory Transformer according to your suggestion, and added further discussion in the paper highlighting their difference in the Related Works section. We summarize the difference here as well:
>
> 1. **Differences between BeT and trajectory transformer**: Firstly, we ran Trajectory Transformer on the Kitchen environment, where it failed to complete any tasks for unconditioned, greedy, or beam search rollouts. We would like to note that this environment is more complicated than the MuJoCo environments (HalfCheetah, Hopper, Walker2d, and Ant) that the paper experimented on and has an order of magnitude fewer samples on the training set.
>
>     While we agree that BeT and Trajectory Transformer based behavior cloning both use some type of discretization to fit demonstration datasets with a minGPT, we believe that is where the similarities end. The primary differences between the algorithms is in our design choices: namely what distributions they model, and consequently how they treat the observations.
>
>     a. **Modeled distribution**: From a provided set of demonstrations, trajectory transformers model the joint distribution P(action, observations). On the other hand, BeT models the conditional distribution P(action | observations). Modeling the joint distribution requires MinGPT to model the forward dynamics of the environment, which can be arbitrarily difficult based on the environment.
>
>     b. **Observation discretization**: Because trajectory transformers have to model the observations as well, it needs to discretize the observation space. As a result, TT cannot extend to high dimensional observational spaces, such as visual observations. This limitation is also acknowledged by the authors of Trajectory Transformers. BeT, on the other hand, does not model the observations and thus does not need to discretize them. Thus BeT can scale to arbitrarily high dimensional observations, as we show in the CARLA environment experiments, where BeT learns behaviors from high dimensional visual observations.
>
>     c. **Efficient historical encoding**: Trajectory transformer encodes each (state, action) pair into a total of $|S| + |A|$ input/output tokens, while BeT encodes them into one input/output token. On a base MinGPT implementation that means a $O( (|S| + |A|)^2 )$ efficiency gain for BeT, or for example 4761x less compute for the same historic context in the Kitchen environment.
>
>     We have added a summary of the above discussions in the appendix to better elucidate the differences for our readers.
>
> 2. **Limitations of k-means**: As you have already gathered, the k-means discretization can be insufficient for binning the actions for very small values of $k$, and we have observed this for $k=1$ or $3$. Our algorithm implicitly assumes a lower variance of actions per bin, which we ensure by using a Bayesian GMM to pick our hyperparameter k. Additionally, there has recently been advances on task-aware action discretizations, such as AQuaDem [1] presented in ICML 2022, which could be worthwhile for more complicated action distributions.
>
> I hope the discussion above answers your questions. We thank you once again for the extremely valuable feedback.
>
> [1] Dadashi, Robert, et al. "Continuous Control with Action Quantization from Demonstrations." arXiv preprint arXiv:2110.10149 (2021).

---

> > ### Comment · Reviewer_xCyV · 2022-08-08
> > **Re Response.**
> >
> > Thank you for the detailed response. My concern about similarities to Trajectory Transformer has been addressed, and I'll raise my score to a 7.

---

### Official Review · Reviewer_4yjG · 2022-07-10

**Rating:** 6
**Confidence:** 4
**Soundness:** 3 good
**Presentation:** 2 fair
**Contribution:** 2 fair

**Summary:**

This paper tackles the problem of behavioral cloning from distributionally multi-modal experience. The proposed solution is "Behavior Transformers" (BeT). Specifically, the authors propose clustering continuous actions into discrete bins using k-means, and model multi-modal policy as categorical distributions with a transformer-based sequence model.

Empirical evaluations are performed on five environments ranging from simple diagnostic point mass to high-dimensional robotics and self-driving environments. Comprehensive analysis is provided to prove that BeT is indeed capable of learning multi-modal behavior in these environment and better than the baseline methods chosen by the authors. Detailed ablations studies are presented to analyzed the importance of each components of the proposed methods.

**Questions:**

* Table 3 shows average BeT performance at the task. Can you be more specific about what they are averaged across?
* The text in the pdf cannot be searched and selected. This makes review difficult.
* Is this a related work? Hausman, K., Chebotar, Y., Schaal, S., Sukhatme, G. and Lim, J.J., 2017. Multi-modal imitation learning from unstructured demonstrations using generative adversarial nets. Advances in neural information processing systems, 30.

**Limitations:**

Very little to no discussion on limitations and potential negative societal impact. Please add relevant pieces to the conclusion section.

**Strengths And Weaknesses:**

Strengths
* Very clear problem statement about imitating multi-modal experience (Sec. 2) and issues with canonical BC that assumes an unimodal expert.
* Clear empirical evidence that BeT is able to imitate multi-modal behavior (Sec. 3.3)
* Use diverse testing environments ranging from simple point mass to high-dimensional robotics and self-driving environments.
* Sufficient ablations that analyze the importance of describe binning, action offsets, interaction history, and transformer architecture.

Weaknesses:
* Imitating multi-modal experience is well-acknowledged problem. However, how the proposed method different from previous contributions in this line is barely discussed. Furthermore, I think there should be citation and discussion about prior efforts in introduction. Intended or not, it somewhat gives me an impression as if this is the first work tackling multimodal imitation learning.
* Insufficient empirical comparison: The only previous contribution about multimodal imitation learning that this work compare with is "Implicit Behavioral Cloning" (IBC) [22]. Because of the IBC paper shows outperforming Mixture Density Network (MDN) [7], a well-established baseline for multi-modal imitation learning, the authors skipped comparing with MDN. These naturally trigger the following questions: (1) Have you compared to MDN or other approaches using mixture distributions apple-to-apple, using the same transformer and also historical context? (2) The exact same question for the comparison between BeT and IBC. Both MDN and IBC can be more easily implemented than the proposed clustering and binning and trained end-to-end. Without fair comparisons, I don't feel convinced about using the proposed method.

---

> ### Author Response · Authors · 2022-08-02
> **Response to Official Review of Paper729 by Reviewer 4yjG**
>
> We thank you for your insightful comments on our paper. To address your concerns, we have run several additional baselines and ablations, and provide clarifications below.
>
> 1. **Proper discussion of prior work in learning from multi-modal experiences**: According to your suggestion, we have added an exposition to learning from multi-modal experiences in the introduction and elaborated the relevant section in Related Work further. Currently, the SOTA here revolves around learning goal-conditioned models. However, multi-modal behavior policies are orthogonal to goal-conditioned learning, as we can easily imagine multiple ways of achieving the same goal. We believe that the scope of multi-modal imitation is wider than learning multi-modal behavior policy learning, and BeT mostly focuses on the latter. Nonetheless, this discussion will aid the reader to understand the landscape of the field.
>
> 2. **Further empirical comparison with transformer-based MDN and IBC**: As per your review, we ran our MDN and IBC baselines with the transformer backbone and historical context.
>
>     * We found the MDN with transformer architecture to be a strong baseline, with stronger performance than MDN with MLP as presented in [1]. However, it still underperforms the transformer with k-means binning, for example by about 15-17% on the Kitchen and Block Push environments, and by 70% on the CARLA environment.
>     * Adding historic context via a minGPT-style encoder significantly slowed down the optimization of the IBC model – to the point of no convergence after 72 hours of training on the Kitchen environment. Note that the IBC results we presented in the paper still takes historical context into account through frame-stacking, and takes about 30 hours to converge on the same environment and the same hardware. BeT on the same dataset and same hardware takes less than an hour to train.
>
> 3. **Clarification of metric**:  On table 3, we report our results normalized by the average number of task successes of standard BeT. Task success is defined per-task, such as reaching the end goal in CARLA, pushing some blocks to target in Block Push, and completing different tasks in the kitchen environment.
>
> 4. **Implementation Complexity**: While we find that any attempts at comparison between two implementations are subjective at best, we would like to point out that beyond boilerplate code added to create ablations and comparison between baselines, our core method is implemented in less than 350 lines in the supplementary submission, split into the implementation of k-means discretizer and the minGPT backbone. On the other hand, the complexity in IBC implementation comes not in the model definition, but in action sampling, namely DFO and Langevin sampling methods, which are implemented in around 400 lines in the official IBC repo.
>
>     At the end of the day, we expect such complexity to be abstracted away from the end user in the format of libraries, and we tried our best to organize and polish our released code, as well as to make our methods fast and memory-lean to enable easier downstream improvements. Additionally, our model is computationally light, with between $10^4$-$10^6$ parameters for the tasks presented here and can be trained to completion in around an hour on a single desktop GPU, which should also positively contribute to downstream research velocity.
>
> 5. **Suggested related work**: We thank the reviewer for the suggestion, and have added it to the related work section. However, we would like to note that [3] relies on continued interaction with the environment in training time following the imitation learning framework introduced by GAIL [4].
>
>
> 6. **Formatting issues**: We have updated the paper to be selectable and searchable. We apologize for any inconveniences caused by this.
>
> In summary, we hope that our clarifications help address some of your concerns around our work. We have made every effort to modify the text in the paper to make these points clear. In light of our clarifications above, we would like to politely ask if you are willing to increase your score assuming we have addressed your concerns. Otherwise, please let us know if you have additional questions.
>
> [1] Florence, Pete, et al. "Implicit behavioral cloning." Conference on Robot Learning. PMLR, 2022.
> [2] Official IBC implementation, https://github.com/google-research/ibc#run-train-eval
> [3] Hausman, K., Chebotar, Y., Schaal, S., Sukhatme, G. and Lim, J.J., 2017. Multi-modal imitation learning from unstructured demonstrations using generative adversarial nets. Advances in neural information processing systems, 30.
> [4] Ho, Jonathan, and Stefano Ermon. "Generative adversarial imitation learning." Advances in neural information processing systems 29 (2016).

---

> > ### Comment · Reviewer_4yjG · 2022-08-09
> > **Re: Author Response**
> >
> > Thanks for performing additional experiments. I would still like to see IBC results at least in this discussion, either intermediate results after 72 hours or on tasks other than Kitchen or both. Also, what is your assumption for why IBC + minGPT converges slower?
> >
> > I would like to echo reviewer oVB2's question "In Figure 5, BeT with history seems to prefer the red mode over the other two modes. While in the dataset all three modes are equally likely. It is unclear, if the BeT model can capture the true underlying distribution of expert demonstrations?", which did not seem to be answered in your response.
> >
> > I think the **limitations** are not really discussed in Sec. 5. I recommend expanding that. Also, the authors chose to not discuss **societal impacts**. Is there a reason?

---

> > > ### Author Response · Authors · 2022-08-09
> > > **Response to Further Questions by Reviewer 4yjG**
> > >
> > > First of all, we thank you for your follow-up comments – we are happy to provide more context to the best of our abilities. Our responses are as follows:
> > > 1. **IBC + MinGPT:**
> > > a. **Success rate:** The IBC + MinGPT model completes 0 tasks after 72 hours of training on both Kitchen and Block Pushing tasks.
> > >  b. **Reason behind slow convergence:** We hypothesize two reasons for the slow convergence:
> > >     * Firstly, the InfoNCE objective in IBC results in training the transformer with a contrastive loss that is much slower to converge compared to the supervised Cross-Entropy or MSE loss.
> > >     * Secondly, with a historical context h(s), the IBC model trains the energy E(a, h(s)) or E(a, s, h(s)). The historical context h(s) starts untrained, which creates a chicken and egg problem where improving h(s) might make the old energy estimate E(a, h(s)) less useful.
> > >     This problem is not as severe when using conv. nets to encode images in the IBC paper because of the deep image prior [1] which shows even an untrained convnet can be useful to extract and restore image information. We see a similar effect where we use a ImageNet-pretrained ResNet representation on our quite off-domain CARLA task and still get positive results. No such useful prior is known about untrained attention-based models.
> > > 2. **Capturing expert demonstrations:** We believe, and reviewer oVB2 seems to agree, that our response satisfactorily answers this question; but to recap (with added emphasis):
> > >     > As you can see from table 2 and figure 4 in our paper, BeT rollouts capture multiple modes in the demonstration distributions, and approximate the demonstration distribution better with a higher entropy (table 2) than any of the baselines. However, **BeT does not perfectly imitate the state distribution in the demonstrations**, which is why you may see phenomena like table 2 or figures 4 and 5 where the demonstration distribution is slightly different from the BeT rollout distribution. We believe perfectly capturing demonstration distribution with a BeT could be an interesting research question, of which our paper is only laying the foundation.
> > >
> > >     We hope this explains the relationship between expert demonstrations’ distribution and BeT rollout distribution better.
> > > 3. **Limitations and Societal Impacts:** Thank you for pointing out the limited discussion about societal impacts. Given the early stage research this work addresses, we do not foresee significant societal impacts. In our commitment to open science, we are open-sourcing all of our code and data, which can help in understanding potential societal impacts in the future. Regarding limitations, we believe that the two significant limitations of our current work, and potential directions for future research, are the lack of real robot results and the lack of conditional behavior generation with deep RL. Both of these limitations have already been mentioned in Sec. 5.
> > > Are there other limitations you would like us to add in this section?
> > >
> > > We hope that our clarifications above have resolved your remaining concerns. In light of our discussion, we would like to politely ask if you are willing to increase your score.
> > >
> > >
> > > [1] Ulyanov, Dmitry, Andrea Vedaldi, and Victor Lempitsky. "Deep image prior." Proceedings of the IEEE conference on computer vision and pattern recognition. 2018.

---

### Official Review · Reviewer_oVB2 · 2022-07-11

**Rating:** 6
**Confidence:** 4
**Soundness:** 3 good
**Presentation:** 3 good
**Contribution:** 3 good

**Summary:**

This paper proposes an imitation learning method to take into account the variance in expert demonstrations. In detail, a novel Behaviour Transformer model is proposed, which uses a k-means clustering based pre-processing approach to extract k modes from the noisy expert demonstrations. The proposed approach exploits the ability of transformer based approaches to predict complex multi-modal distributions.  The proposed model is evaluated on diverse tasks such as, CARLA, Block Pushing and Franka Kitchen.

**Questions:**

In addition to the points above,

* The motivation behind the choices of the baselines should be made clear? Currently the baselines chosen are quite simple.

* It is unclear if the method can truly deal the variance in expert demonstrations, expect for the simplistic point mass environments?

* Why does the VAE baseline fail completely on the CARLA experiments?

**Limitations:**

Currently the paper does not discuss the limitations of the the proposed BeT model in detail, e.g. wrt the computational resources required for training, whether the proposed method scales to large scale datasets and tasks, the suitable application areas.

**Strengths And Weaknesses:**

Strengths,
* The proposed method is simple and is able to capture the variance of expert demonstrations as demonstrated in case of the point mass environments.
* The paper is well written and easy to understand.
* The paper includes ablations on the effect of discrete binning, action offsets and historical contexts, which shows the importance of these components.


Weaknesses,

* Effect of L2 loss in L130: Even after k-means discretization in L106, there could be considerable variance within each cluster. Using the L2 loss as in L130 would kill this variance? In case of the point mass environment in Figure 2 and 5, this loss would not have a large effect as the within cluster variance is low. The paper should include examples where the true variance of the expert demonstrations is captured.

* In Figure 5, BeT with history seems to prefer the red mode over the other two modes. While in the dataset all three modes are equally likely. It is unclear, if the BeT model can capture the true underlying distribution of expert demonstrations?

* Importance of the transformer model: While there is not denying the effectiveness of transform models, for the simple tasks considered here, e.g. point mass environments, it is not clear if the MinGPT model is actually necessary. It would be interesting to consider a LSTM based baseline as it would enable the modelling of historical contexts unlike MLPs considered in Table 3.

* The CARLA experimental setting seems very simple. It is also unclear how the expert demonstrations are captured? The amount of variance (if any) in the expert demonstrations is unclear. This is further illustrated by the fact that RBC baseline which cannot deal with variance in the expert demonstrations works so well. Moreover, the paper should include further details like number of traffic participants in the scene, so that the reader can correctly judge the complexity of the task.

* Baselines: The paper compares to only simple baselines like VAE, Flow and IBC. Prior state of the art works such as GAIL [a] are not considered as baselines.

* Equation numbers are missing, e.g. L130.

[a] Generative Adversarial Imitation Learning, Ho et. al.

---

> ### Author Response · Authors · 2022-08-02
> **Response to Official Review of Paper729 by Reviewer oVB2 [2/2]**
>
>
> 5. **Failure of VAE in CARLA**:
>     > Why does the VAE baseline fail completely on the CARLA experiments?
>
>     The CARLA experiments are our longest time-horizon experiments in the paper (with more than 550 timesteps, as opposed to around 100 timesteps on Block push or 280 timesteps on Kitchen). Thus any method with significant high-frequency action noise (in this case, VAE/flow/IBC) struggles to perform well, since the success/failure metric is binary and going out of distribution in just one out of 550+ timesteps is enough to make them fail.
>
>
> In light of the above clarifications, assuming we have addressed your concerns, we would like to politely ask if you are willing to increase your score. Otherwise, please let us know if you have additional questions and we would be more than happy to discuss further.
>
> [1] Pertsch, Karl, Youngwoon Lee, and Joseph Lim. "Accelerating Reinforcement Learning with Learned Skill Priors." Conference on Robot Learning. PMLR, 2021.
> [2] Singh, Avi, et al. "Parrot: Data-Driven Behavioral Priors for Reinforcement Learning." International Conference on Learning Representations. 2020.
> [3] Florence, Pete, et al. "Implicit behavioral cloning." Conference on Robot Learning. PMLR, 2022.

---

> > ### Comment · Reviewer_oVB2 · 2022-08-08
> > **Post Response**
> >
> > The rebuttal has addressed most of my concerns. I have raised my score accordingly. The only remaining concern of significance is that the experimental setting is, in general, quite simple, e.g. on CARLA, and quite far removed from the complex scenarios, e.g. with respect to the routes and traffic density in CARLA, compared to the SOTA methods. Nevertheless, the paper does make a solid contribution.

---

> ### Author Response · Authors · 2022-08-02
> **Response to Official Review of Paper729 by Reviewer oVB2 [1/2]**
>
> Thank you for your insightful comments and constructive feedback on our paper. We are glad that you find our work simple, and easy to understand. To address your concerns, we have run several additional baselines and ablations, and provide clarifications below.
>
> 1. **Variance captured by BeT**:
>
>     > It is unclear if the method can truly deal the variance in expert demonstrations, expect for the simplistic point mass environments… It is unclear, if the BeT model can capture the true underlying distribution of expert demonstrations?
>
>     We would like to point to the diversity of modes covered by BeT in complex environments beyond point-mass. As you can see from table 2 and figure 4 in our paper, BeT rollouts capture multiple modes in the demonstration distributions, and approximate the demonstration distribution better with a higher entropy (table 2) than any of the baselines. However, BeT does not perfectly imitate the state distribution in the demonstrations, which is why you may see phenomena like table 2 or figures 4 and 5 where the demonstration distribution is slightly different from the BeT rollout distribution. We believe perfectly capturing demonstration distribution with a BeT could be an interesting research question, of which our paper is only laying the foundation.
>
>     As you suggested in your first point:
>
>     > Even after k-means discretization in L106, there could be considerable variance within each cluster.
>
>     k is a hyperparameter of our algorithm, and the Bayesian GMM algorithm that we used to choose k specifically tries to reduce the variance within each cluster. Further, once we have chosen a k, you are correct in identifying that we make an assumption in using L2 loss for offsets. Namely, we assume that **once the conditioning state is fixed**, the action distribution assigned to a bin is unimodal. Note that this still allows the action distribution in a bin to be multimodal while conditioned by different states. With our assumption, we reduce the problem of multimodal action prediction into multimodal bin prediction and unimodal action prediction over such bins.
>
>
> 2. **Importance of Transformer**: Per your feedback, we added new baselines to our paper (Table 3) with different historical context models, namely LSTMs and Temporal convolution based models. In our experiments, we see that they generally get vastly outperformed by Transformers. Additionally, in our experience they are much harder to train stably. In particular, LSTM-based model performance is less than 5% of minGPT-based model performance in all cases, while Temporal Convolution has approx. 75% of minGPT performance in the best case. Please see table 3 for detailed results.
>
>
> 3. **Motivation behind baselines**:
>
>     > Prior state of the art works such as GAIL [a] are not considered as baselines… The motivation behind the choices of the baselines should be made clear
>
>     While GAIL is an important work in imitation learning, it is not compatible with our setting since it learns by interacting with the environment while training. On the other hand, we only consider and compare behavioral cloning algorithms that learn from offline demonstration data without any further interactions with the environment. We chose our baselines (VAE, Flow, and IBC) inspired by state-of-the-art behavior learning algorithms respectively [1, 2, 3] that use state-of-the-art generative model based architectures capable of learning multi-modal behaviors from offline demonstration data. Additionally, we create further baselines by ablating and studying different parts of our algorithm. We hope that you find the motivation behind our baselines clear with this explanation, and find that they reflect the state-of-the-art in learning from multi-modal demonstrations.
>
>
> 4. **CARLA environment**: Per your request, we have added further details about the CARLA environment in the appendix. To summarize, we generate demonstrations by a standard trajectory-following agent with some injected noise in the actions of the demonstrators. The variance in the demonstrations come mostly from the bi-modal distributions of path taken, and the injected noise in the actions.
>
>     We intentionally keep the CARLA environment simple with no traffic participants because we wanted this environment to be the simplest image-based step beyond the point-mass environments, showing BeT behaviors with two very clear modes of demonstrations.

---

### Author Response · Authors · 2022-08-02
**Global comments and revision summary in response to the reviews of Paper729**

We thank the reviewers for your insightful and constructive feedback. We are glad that this work was received positively, although some important concerns were raised. To address these concerns, we have run **all** the additional baselines that were requested. While a detailed response to your feedback is available in individual replies to your review, a summary of our revision is presented below:
1. **Comparison with more baselines and ablations:** We have performed the following baselines and ablations experiments:
    1. Comparisons with transformer-based MDN (reviewer 4yjG). This baseline performs better than the MLP-based MDN presented in IBC [1], but still falls short of BeT performance, for example by 14% in the Kitchen environment or 70% in CARLA.
    2. Transformer-based IBC (reviewer 4yjG), which unfortunately slows down convergence to the point where we trained for 72 hours and yet could not converge to a local minima.
    3. Uniform quantization head instead of k-means (reviewer uHYM), which performs competitively with other baselines but falls behind BeT by 4-10% across environments.
    4. Trajectory transformers sans-rewards (reviewer xCyV), which required a $O((|A| + |S|)^2)$ blowup in computational cost on the straightforward implementation (for example, 4761x more compute on Kitchen) and failed to complete any tasks on the Kitchen environment.
    5. Alternative historical context models, such as LSTM and Temporal Convolutions (reviewer oVB2), which were much more unstable to train and perform significantly worse than transformer-based models in all environments. In particular, LSTM-based models’ performance is below 5% of minGPT-BeT models in all cases, while Temporal Convolution has approx. 75% of minGPT performance in the best case.

    With our new experiments, we see a general trend that transformer based models outperform other architectures, such as MLPs, LSTMs, or temporal convolutions. Importantly, at the same time, the presented k-means discretizer based BeT outperforms all other discretization or multi-modal prediction methods such as uniform quantizations or MDN. We have added the results from these experiments in the ablation section and the appendix.

2. **Additional discussion on prior works:** Addressing comments by reviewer 4yjG and reviewer xCyV, we added further discussion about prior approaches in learning from multi-modal demonstrations, and how our approach in learning a single multi-modal policy differs from them. Specifically, we differentiate between the standard goal-conditioned policy learning approach from our approach, and discuss how the distribution modeled in BeT is different and conceptually simpler than preceding works.

3. **Added environment details:** Per reviewer oVB2’s comments, we added further details about the CARLA self-driving environment in the appendix section, specifically about the variance in the demonstrations and existence of distractors.

4. **Updated details on computational considerations:** As suggested by reviewer uHYM, we updated our computational considerations section to highlight both training and evaluation running times and efficiency gains compared to baselines. In general, we have found that during rollouts, our model can be 10-25x faster than IBC, and 300x faster than Trajectory Transformers.

5. **Formatting updates:** As suggested by reviewer oVB2, we added equation numbers, and made the PDF selectable and searchable per the suggestion of 4yjG.

We hope that these updates to our paper inspire further confidence in our work. At the same time, we invite any further questions or feedback that you may have on this paper.

[1] Florence, Pete, et al. "Implicit behavioral cloning." Conference on Robot Learning. PMLR, 2022.

---

### Author Response · Authors · 2022-08-08
**A note to reviewers**

Dear reviewers,
We would like to invite you once again to share any follow up questions that you may have following our rebuttal, so that we can respond to them before the quickly approaching author response period. Otherwise, assuming we have responded to your concerns satisfactorily through our updates to the paper and our comments, we would like to ask you if you are willing to increase your review scores.
Regards,
Authors of "Behavior Transformers: Cloning $k$ modes with one stone"

---

### Meta-Review · Area_Chair_P7rD · 2022-08-26

**Recommendation:** Accept
**Confidence:** Certain

**Metareview:**

*Summary*

The paper addresses the problem of learning from expert demonstrations, focusing on the setting where the demonstrations are pre-collected, rewards are absent, and the distribution of demonstration trajectories contains multiple modes (limiting the performance of behavior cloning). The proposed approach uses k-means to cluster continuous actions into discrete tokens which are modeled using a transformer architecture with an additional continuous offset from the cluster centers.

*Reviews*

The discussion has resolved all reviewer concerns and strengthened the paper with additional baselines, ablations, and positioning relative to related work. All four reviewers are in agreement that this approach is novel, well justified, effective in multiple empirical settings, and that the ablation studies clearly establish why the model performs well. The final reviewer ratings are 6 (WA), 6 (WA), 7 (A) and 8 (SA). At least two of the reviewers see this paper as high impact, and I agree. I therefore recommend this submission can be accepted and considered for an outstanding paper award.

*Potential Impact*

In generative image modeling, the shift to modeling the continuous space of pixels using discrete tokens and transformers helped underpin recent massive improvements in quality (e.g., in the [DALL-E](https://arxiv.org/pdf/2102.12092.pdf) paper and others). In the image domain, the two-stage approach (learning a codebook, then training a transformer) seems to successfully capture both high-frequency details and low-frequency structure. Therefore it's interesting to see this paper apply similar ideas to the modeling of behavior/actions. To my knowledge this is the first paper to open up this direction and it could lead to further advancements with the application of more advanced clustering / codebook learning techniques and so on (as has occurred in the image domain). Therefore I see the potential impact of this paper as high.

**Award:**

Yes

---

### Decision · Program_Chairs · 2022-09-14

Accept